# ON THE ROBUSTNESS OF SELF-SUPERVISED REPRESENTATIONS FOR SPOKEN LANGUAGE MODELING

## ABSTRACT

Self-supervised representations have been extensively studied for discriminative and generative tasks. However, their robustness capabilities have not been extensively investigated. This work focuses on self-supervised representations for spoken generative language models. First, we empirically demonstrate how current state-of-the-art speech representation models lack robustness to basic signal variations that do not alter the spoken information. To overcome this, we propose an effective and efficient method to learn robust self-supervised speech representation for generative spoken language modeling. The proposed approach is based on applying a set of signal transformations to the speech signal and optimizing the model using an iterative pseudo-labeling scheme. Our method significantly improves over the evaluated baselines when considering encoding metrics. We additionally evaluate our method on the speech-to-speech translation task. We consider Spanish-English and French-English conversions and empirically demonstrate the benefits of following the proposed approach.

## 1 INTRODUCTION

Self-supervised speech models were shown to learn effective representations for various downstream tasks (Hsu et al., 2021; Chen et al., 2022; Baevski et al., 2020). These models were mainly evaluated on discriminate tasks, such as automatic speech recognition, speaker verification, intent classification, etc. (Yang et al., 2021). Recently, Lakhotia et al. (2021) demonstrated that such self-supervised learning (SSL) representations can be used for Generative Spoken Language Modeling.

Generative Spoken Language Modeling (GSLM) is the task of learning the acoustic and linguistic characteristics of a language from raw audio. In other words, a discrete representation of the audio signal is being learned. Then a speech-language model is trained on top of the obtained representation. Finally, a neural vocoder converts the output tokens to raw audio. As the discrete speech representation often operates over tokens extracted every twenty milliseconds of audio, sequences can be long and contain repetitions, e.g., `10 11 11 11 21 32 32 32 21`. Preliminary studies have found that removing sequential repetitions of units improves performance, hence applying it universally (Lakhotia et al., 2021). For example, a pseudo-text `10 11 11 11 21 32 32 32 21` becomes `10 11 21 32 21`. This framework was shown to be effective in modeling multiple levels of the speech utterance, namely prosody, and content (Lakhotia et al., 2021; Kharitonov et al., 2021a; Borsos et al., 2022), speech codec (Polyak et al., 2021), speech emotion conversion (Kreuk et al., 2021), spoken dialogue (Nguyen et al., 2022), and speech-to-speech translation (Lee et al., 2021a; Popuri et al., 2022; Lee et al., 2021b).

An essential prerequisite for such an audio representation to be used in real-world conditions is robustness to various signal corruptions. Although the aforementioned audio representation models have shown effectiveness in many tasks, they were mainly evaluated on academic benchmarks.

In this work, we evaluate current state-of-the-art self-supervised speech representation models on what are arguably the most basic signal variations, namely time-stretch, pitch-shift, additive-noise, and reverberation. Our premise is that while these variations modify the signal, its' underlying content remains the same, especially under the tokens repetition removal process. Therefore, a robust representation should be affected by such variations to a minimal extent.

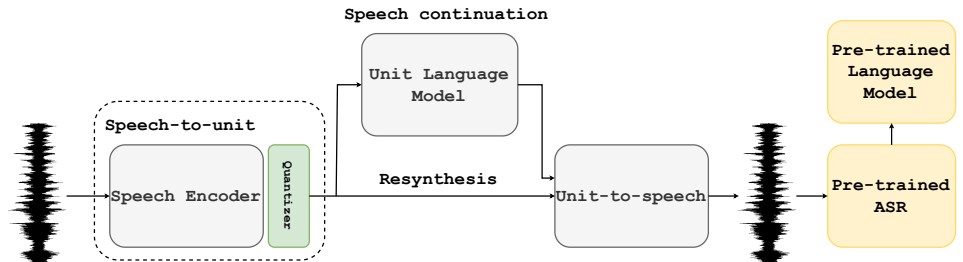

Figure 1: Generative Spoken Language Modeling is composed of three components: (i) Speech-to-unit, (ii) Unit language model, and (iii) Unit-to-speech. Pre-trained ASR and language models are used to evaluate those components.

As a first step, we propose a set of metrics for evaluating the model's robustness. Then, we point to the lack of robustness of these models with respect to the aforementioned variations. Next, we design a simple and effective method for learning robust discrete representation on top of any speech SSL model. We demonstrate how such a method greatly improves robustness. Then, we empirically show that performance improves on several tasks for various SSL models. Specifically, we evaluate the newly proposed speech encoders when considering established encoding metrics, i.e., ABX, WUGGY, BLIMP (Nguyen et al., 2020), together with a high-level downstream task in the form of speech-to-speech translation.

## 2 BACKGROUND

The general Generative Spoken Language Modeling (GSLM) pipeline is comprised of three main modules: (i) Speech-to-unit, (ii) Unit language model, and (iii) Unit-to-speech, where each of these modules is trained separately. Speech resynthesis can be achieved while ignoring the language model and directly feeding the quantized units into the unit-to-speech module (Polyak et al., 2021) (See Figure 1 for a visual description of the overall pipeline). In the following paragraphs, we give detailed background for each of the three components mentioned above, including the standard evaluation methods.

**Speech-to-unit** module encodes the raw speech signal into a discrete representation. The common approach is first to encode the speech into a continuous representation and then quantize the representation to achieve a sequence of discrete units (Lakhotia et al., 2021; Polyak et al., 2021; Popuri et al., 2022; Lee et al., 2021a; Kharitonov et al., 2021a; Kreuk et al., 2021; Kharitonov et al., 2022; Nguyen et al., 2022; Borsos et al., 2022; Tjandra et al., 2019; 2020).

Formally, denote the domain of audio samples by $\mathcal{X} \subset \mathbb{R}$. The representation for a raw signal is therefore a sequence of samples $x = (x_1, \ldots, x_T)$, where $x_t \in \mathcal{X}$ for all $1 \leq t \leq T$.

Consider an encoder network, $f$, that gets as input the speech utterance and outputs a sequence of spectral representations sampled at a low frequency as follows $f(x) = (v_1, \ldots, v_{T'})$. Note that we do not assume anything about the structure of the encoder network $f$. Lakhotia et al. (2021), evaluated several speech encoders, namely, Mel-spectrogram, Contrastive Predictive Coding (CPC) (Oord et al., 2018), wav2vec2 (Baevski et al., 2020), and HuBERT (Hsu et al., 2021).

Since the representations learned by such models are usually continuous, a k-means algorithm is applied over the models' outputs to generate discrete units, denoted as $z = (z_1, \ldots, z_{T'})$. Each element $z_i$ in $z$ is a positive integer, $z_i \in \{1, .., K\}$ for $1 \leq i \leq T'$, where $K$ is the number of discrete units. We denote the quantization model with $E$.

**Unit Language Model** is trained on the extracted discrete units, $z$. Such a language model learns a probability distribution of the learned unit sequences, which enables direct modeling of speech data without textual supervision.

The language model can be used to generate speech conditionally or unconditionally, replicating what toddlers achieve before learning to read. Moreover, such a modeling framework allows for capturing and modeling prosodic features (Kharitonov et al., 2021a), as well as speaker identity (Borsos et al., 2022), or even natural dialogues (Nguyen et al., 2022). This is in contrast to using textual features, as they do not encode such information.

**Unit-to-speech** module converts the speech discrete units to a raw waveform. Lakhotia et al. (2021) used a Tacotron2.0 (Shen et al., 2018) based model followed by WaveGlow (Prenger et al., 2019) vocoder. Later, Polyak et al. (2021) proposed a unit-based vocoder based on the HiFi-GAN architecture to convert units to speech directly. Such a paradigm seems to provide high-quality generations with better efficiency as it uses only one model rather than two. Kreuk et al. (2021) and Lee et al. (2021a) additionally improved the unit based vocoder to include emotional tokens for speech emotion conversion tasks, and duration modeling for direct speech-to-speech translation.

**Evaluation metrics.** Evaluating such a complex pipeline comprised of several components is a challenging task. Lakhotia et al. (2021) proposed a set of evaluation metrics aiming for each of the modules. Overall the proposed metrics can be divided into four main groups: (i) acoustic encoding using ABX, bitrat, (ii) language encoding using WUGGY, BLIMP (Nguyen et al., 2020; Lakhotia et al., 2021), (iii) resynthesis using Phoneme/Word Error Rate; (iv) speech generation using VERT (Lakhotia et al., 2021), Meaningfulness Mean Opinion Score.

## 3 ROBUSTNESS OF SELF-SUPERVISED MODELS

The first step toward developing an effective spoken language model is to develop a robust representation. The focus of a robust representation should be on the spoken information rather than unrelated signals, such as background noise or reverberations. In the following section, we propose a metric for quantifying the degree to which augmentations change the resulting encoding.

### 3.1 UED SCORE

A spoken language model is built on top of a discrete representation of a continuous encoder. We examine the robustness of the discrete space to augmentations that do not change the spoken content. Therefore, we are interested in a sequential distance metric between two discrete representations. It is essential to note that augmentations can alter the spatial dimension of the signal. For example, stretching a signal results in more frames, yielding a longer representation sequence. Hence, the metric should be able to measure the distance between two sequences of different lengths. Ideally, it will consider the number of deletions, insertions, and substitutions that occur due to augmenting the input data. For this purpose, we find the Levenshtein distance a good fit (Levenshtein, 1966). The Levenshtein distance measures the minimum changes one should make to modify one sequence to another. It has two essential properties: the first is that the score is non-negative, and when the sequences are equal, the metric equals zero. The second property is that the maximum value it can get equals the longer sequence length between the two sequences. We provide a detailed explanation of the Levenshtein distance in the Appendix material.

We aggregate the distance values over the evaluation set while considering the sequence length. This is desirable since we want to normalize scores for sequences in different lengths, and the Levenshtein distance's maximum value is the original sequence's length. Another essential property of a spatial metric is repetitions. Consider time stretch as an example, it changes the number of the input frames, but one would expect the deduplicated quantized signal to be the same as before the augmentation. Hypothetically, one can maximize the score by stretching the signal infinitely. To eliminate such dependencies, we compute the score on a deduplicated quantized representation. Formally, our final metric is:

**Definition 3.1** (Unit Edit Distance). *Given a continuous encoder $f : \mathbb{R}^T \to \mathbb{R}^{T'}$, a quantizer $E : \mathbb{R}^{T'} \to \{1, .., K\}^{T'}$, and an input augmentation $g : \mathbb{R}^{T'} \to \mathbb{R}^{\widehat{T'}}$. The deduplicated unit edit distance on the evaluation set $\mathcal{D}$ is:*

$$UED_{\mathcal{D}}(E, f, g) \triangleq \sum_{x \in \mathcal{D}} \frac{1}{T'_x} LEV\left((E \circ f)(x), (E \circ f \circ g)(x)\right), \qquad (1)$$

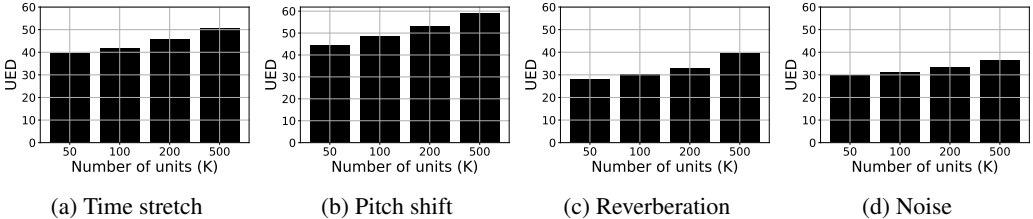

Figure 2: UED scores for various augmentations and number of clusters. We note that the UED is relatively high (the distance is normalized). We also note that the UED monotonically increases with the number of units used. We multiply the scores by a hundred.

where $T'_x$ is the number of frames of a sample $x$.

Ideally, a perfect spoken language quantizer obtains a zero distance after deduplication. Next, we study state-of-the-art spoken language representations using our proposed metric in different settings.

## 3.2 EVALUATION

In the following, we study current state-of-the-art representations for generative spoken language modeling using the proposed metric. The current popular quantization technique is a k-means model trained on top of a pre-trained encoder (Lakhotia et al., 2021). In our evaluation setup, we use a different number of clusters and encoder architectures. Our ablation study include quantizers with 50, 100, 200, and 500 clusters. We further investigate our metric on top of HuBERT (Hsu et al., 2021), wav2vec2 (Baevski et al., 2020), and WavLM (Chen et al., 2022). For readability, throughout the paper, we report results for the HuBERT model while leaving the rest of the results in the Appendix material.

### 3.2.1 AUGMENTATIONS

This work focus on four simple signal modifications which mimic real-world signal variations:

**Time stretch.** We use the Phase Vocoder method (Karrer et al., 2006) to stretch or shrink the time domain signal with a rate of $\tau$ without changing the pitch. For example, $\tau = 1.2$ speeds up the signal by $20\%$.

**Pitch shift.** We change the original pitch of the speech signal by a given number of semitones using the resampling method over the time-stretched signal (Karrer et al., 2006).

**Reverberation.** We follow a similar setting of Chazan et al. (2021), in which we consider an Acoustic Transfer Function (ATF) to be simulated using the pyroomacoustics (Scheibler et al., 2018) audio room simulations package. We randomly sample room dimensions, microphone location, and source location, then convolve the ATF with the speech signal.

**Noise injection.** We mix a given speech signal with non-stationary additive noise, using a randomly samples Signal-to-Noise Ratio (SNR) in the range of $[0, 5]$. Background noises are sampled from the Deep Noise Suppression (DNS) challenge (Reddy et al., 2020) which includes a diverse set of noise types from AudioSet (Gemmeke et al., 2017), Freesound, [1] and Demand (Thiemann et al., 2013).

### 3.2.2 RESULTS

In Figure 2, we use our metric to study the robustness of k-means trained on top of HuBERT with various augmentations and values of $K$. This evaluation points to the lack of robustness of the current state-of-the-art representation of simple, non-spoken augmentations. For example, for time stretch augmentation, the UED score is between 39 and 51. Considering that UED is computed on deduplicated signals, those numbers are high. Moreover, this number increases as a function of $K$.

---

[1] https://freesound.org/

Figure 3: Illustration of our method: We forward a clean signal through an encoder followed by a pre-trained quantizer (k-means). Next, we forward an augmented signal through the same encoder, followed by a new quantizer (green). The CTC loss between the deduplicated output of the clean signal and the output of the augmented signal is used to learn the parameters of the new quantizer. In the iterative approach, post the convergence of the learned quantizer $E_0$, we freeze it and learn a new quantizer $E_1$ that distills information from $E_0$.

The high numbers and the monotonicity of the UED as a function of $K$ are consistent for all values of $K$, augmentations, and models we experimented with (HuBERT, wav2vec2, and WavLM). Next, we propose a method that improves the robustness of such representations.

## 4 PSEUDO-LABELING FOR ROBUST DISCRETE REPRESENTATION

Our findings in Section 3 suggest that current state-of-the-art representations may be too sensitive to augmentations that do not alter spoken information. Preliminary robustness research focused primarily on noise augmentation. This is convenient since the signal length is not affected by such augmentations. In practice, real-world augmentations may modify the signal length. In order to work with various types of augmentations, we must align the original and augmented sequences. The following section presents a pseudo-labeling, alignment-based approach to learning a robust quantizer.

### 4.1 PSEUDO-LABELING

The GSLM encoding framework comprises a raw audio signal forwarded through an encoder, then a quantizer. The quantizer is learned on top of a trained encoder, e.g., k-means trained on each embedding vector extracted from a given layer in HuBERT.

As discussed above, we do not want to limit the robustness process to a family of augmentations that do not change the signal's length. To align and use augmentations that may modify the signal's length, we use the Connectionist Temporal Classification (CTC) loss (Graves et al., 2006). The CTC operation computes the probability of an alignment based on the predicted and target sequences. Finally, the CTC loss considers the negative log-likelihood produced by the CTC operation.

We forward a clean signal through an encoder $f$ followed by a pre-trained quantizer $E_0$. Parallelly, we forward an augmented signal through the same encoder $f$ and train a non-linear multi-layer-perceptron $E_1$. Using the CTC loss, which accounts for the alignment between the outputs, we learn the parameters of $E_1$. Formally, the probability given by the CTC loss for a single data point $x$ follows

$$\ell(E_0, E_1, x, g) \triangleq -p\left((E_0 \circ f)(x)|(E_1 \circ f \circ g)(x)\right) = -\sum_{\mathcal{A} \in \mathcal{A}_x} \prod_{t=1}^{r} p_t(a_t|(E_1 \circ f \circ g)(x)), \quad (2)$$

where $\mathcal{A}_x$ is the set of all possible alignments. Finally, for a training set $\mathcal{D}$, a set of augmentations $\mathcal{G}$, a pre-trained quantizer $E_0$, and a learned quantizer $E_1$, our loss function is as follows:

$$\mathcal{L}_\mathcal{D}(E_0, E_1, \mathcal{G}) \triangleq \mathbb{E}_{x \sim \mathcal{D}, g \sim \mathcal{U}(\mathcal{G})} \left[\ell(E_0, E_1, x, g)\right]. \quad (3)$$

Note that the alignment between the predicted and target sequences is many-to-one. Thus, one or more output units can be aligned to a single target unit. Hence, to work with augmentations that stretch the signal, we are required to deduplicate the target sequence.

Table 1: Unit edit distance study: Using our metric, we assess the robustness of various quantization methods on top of a HuBERT representation. This study uses four different augmentations: time stretching, pitch shifting, reverberation, and noise injection. The non-iterative (Section 4.1) and iterative (Section 4.2) methods significantly and consistently improve the robustness of k-means. Pseudo-labeling accounts for most of the improvement. By applying our method iteratively, we can improve it further. For readability, we multiply the scores by a hundred.

| # units | Method | Augmentation | | | |
|---|---|---|---|---|---|
| | | Time | Pitch shift | Reverberation | Noise |
| 50 | k-means | $39.61_{\pm 0.37}$ | $44.33_{\pm 0.92}$ | $28.25_{\pm 0.61}$ | $29.74_{\pm 0.31}$ |
| | Ours | $27.91_{\pm 0.42}$ | $30.74_{\pm 0.71}$ | $20.16_{\pm 0.60}$ | $25.33_{\pm 0.36}$ |
| | Ours (Iterative) | $\mathbf{26.89}_{\pm 0.33}$ | $\mathbf{30.22}_{\pm 0.79}$ | $\mathbf{19.89}_{\pm 0.54}$ | $\mathbf{24.67}_{\pm 0.29}$ |
| 100 | k-means | $41.97_{\pm 0.42}$ | $48.68_{\pm 0.96}$ | $30.42_{\pm 0.69}$ | $31.38_{\pm 0.33}$ |
| | Ours | $31.05_{\pm 0.39}$ | $34.77_{\pm 0.92}$ | $22.21_{\pm 0.63}$ | $28.05_{\pm 0.31}$ |
| | Ours (Iterative) | $\mathbf{29.72}_{\pm 0.41}$ | $\mathbf{32.84}_{\pm 0.91}$ | $\mathbf{21.31}_{\pm 0.71}$ | $\mathbf{25.06}_{\pm 0.31}$ |
| 200 | k-means | $45.59_{\pm 0.39}$ | $53.14_{\pm 1.01}$ | $32.89_{\pm 0.72}$ | $33.34_{\pm 0.38}$ |
| | Ours | $34.40_{\pm 0.46}$ | $38.51_{\pm 1.09}$ | $24.10_{\pm 0.66}$ | $30.19_{\pm 0.37}$ |
| | Ours (Iterative) | $\mathbf{32.99}_{\pm 0.42}$ | $\mathbf{36.45}_{\pm 1.03}$ | $\mathbf{22.94}_{\pm 0.67}$ | $\mathbf{26.76}_{\pm 0.31}$ |
| 500 | k-means | $50.60_{\pm 0.42}$ | $58.92_{\pm 0.98}$ | $39.71_{\pm 0.81}$ | $36.47_{\pm 0.44}$ |
| | Ours | $38.04_{\pm 0.44}$ | $43.48_{\pm 1.03}$ | $28.43_{\pm 0.73}$ | $29.99_{\pm 0.45}$ |
| | Ours (Iterative) | $\mathbf{36.50}_{\pm 0.49}$ | $\mathbf{40.82}_{\pm 1.02}$ | $\mathbf{25.78}_{\pm 0.74}$ | $\mathbf{27.51}_{\pm 0.49}$ |

Intuitively, this process distills quantization knowledge from the pre-trained quantizer into the new quantizer while injecting $E_1$ knowledge about the contextual similarity between the original and augmented signals.

A significant advantage of our method is that it is highly efficient. Our method requires training only a relatively small amount of parameters. In contrast to previous methods that train HuBERT from scratch, which takes up to seven days on 32 GPUs, our method converges in a few hours on a single GPU. In fact, our experiments show that learning the parameters of the encoder performs worse than freezing them. The UED metric is boosted, but the ABX scores are negatively affected. The freezing of the upstream model thus serves as a suitable regularizer.

## 4.2 ITERATIVE PSEUDO-LABELING

In the previous section, we presented a pseudo-labeling approach that relies on a converged quantizer $E_0$, e.g., k-means on top of HuBERT. This raises the question of whether it is possible to enhance the robustness of the learned quantizer $E_1$ by iteratively replacing the pre-trained quantizer with the converged quantizer and learning another MLP on top of it. It turns out that such a process can further improve the robustness.

The iterative process begins with a pre-trained quantizer $E_0$, then, as in Section 4.1 we learn a robust quantizer $E_1$. Upon $E_1$ convergence, we replace $E_0$ with $E_1$ and use it as the pre-trained quantizer. Then, we learn a new MLP $E_2$ on top of the converged $E_1$. We repeat this process $K$ times. This process needs more careful training. We note that it is essential to replace the quantizers only post-convergence.

## 5 EXPERIMENTS

In the following, we assess the efficacy of our method using state-of-the-art self-supervised representations and popular discriminative and generative metrics. It is important to note that a single metric cannot tell the whole story. For example, similarly to perplexity, all representations can be assigned to the same cluster to achieve a perfect unit edit distance. In this section, we first examine our proposed method using the unit edit distance along with other discriminative and generative performance metrics. Then, we show that our method further improves downstream tasks.

Table 2: Discriminative and generative evaluation metrics: We evaluate the ABX score on the 'clean' and 'other' development sets from Librispeech. Our method improves the ABX, WUGGY, and BLIMP metrics in all setups.

| # units | Method | ABX (clean) ↓ | | ABX (other) ↓ | | WUGGY ↑ | BLIMP ↑ |
|---|---|---|---|---|---|---|---|
| | | Within | Across | Within | Across | | |
| 50 | k-means | 7.52 | 8.90 | 9.84 | 13.5 | 66.12 | 54.91 |
| | Ours | 6.76 | 7.72 | **9.03** | **11.78** | **67.59** | 55.76 |
| | Ours (Iterative) | **6.63** | **7.55** | 9.53 | 12.14 | 67.42 | **57.04** |
| 100 | k-means | 6.37 | 7.72 | 8.4 | 12.29 | 67.70 | 56.16 |
| | Ours | 5.50 | **6.21** | **7.24** | **10.11** | 67.79 | **57.01** |
| | Ours (Iterative) | **5.39** | 6.22 | 7.46 | 10.20 | **68.20** | 56.99 |
| 200 | k-means | 5.99 | 7.14 | 8.23 | 11.51 | 66.51 | 54.64 |
| | Ours | 5.29 | **6.01** | 7.22 | 9.78 | 70.51 | 56.19 |
| | Ours (Iterative) | **5.19** | **6.00** | **7.18** | **9.70** | **70.68** | **56.26** |
| 500 | k-means | 5.98 | 6.98 | 7.89 | 11.43 | 66.92 | 55.97 |
| | Ours | 5.16 | 6.03 | 7.06 | 9.76 | **70.13** | 55.19 |
| | Ours (Iterative) | **4.96** | **5.73** | **6.93** | **9.63** | 69.33 | **56.93** |

This section begins by describing the experimental setup (Section 5.1). In Section 5.2 we use our proposed metric from Section 3 to analyze the robustness of our method to augmentations. In Section 5.3 we study the discriminative capabilities of our method using the ABX test (Schatz et al., 2013). Then, we evaluate our methods using generative metrics such as WUGGY and BLIMP (Nguyen et al., 2020; Lakhotia et al., 2021). Finally, we demonstrate the effect of using our robust quantizer's units in downstream tasks such as speech-to-speech translation.

## 5.1 EXPERIMENTAL SETUP

**Models.** We study our method using the base versions of HuBERT, wav2vec2, and WavLM. Similar to prior work, for HuBERT and WavLM, we use the ninth and sixth layers for wav2vec2. For readability, we report results for HuBERT in the main paper. The results for wav2vec2 and WavLM are presented in Appendix B. In our quantizer learning process, we use a learning rate of 0.0001, a batch size of 32, and Adam optimizer (Kingma & Ba, 2014). Our quantizer is composed of three fully connected layers with LeakyReLU activation between them. The dimensions of those layers are determined by the division floor of the difference between the upstream dimension to the number of units. We train our quantizer using a single NVIDIA V100 GPU.

**Datasets.** To match the current k-means popular training set, we use the Librispeech-100h to learn our quantizer (Panayotov et al., 2015). We analyze our metric using the 'clean' and 'other' development sets from Librispeech. The augmentations in all setups include time stretch, pitch shift, reverberation, and noise injection (exact parameters are detailed in Section 3.2.1). For the WUGGY and BLIMP evaluations, we use the 'big' transformer language model from Lakhotia et al. (2021).

## 5.2 ROBUSTNESS

In Section 3, we presented an evaluation metric that assesses the robustness of a quantized speech representation to augmentations. The metric is insensitive to changes in the length of the signal. Using it, we investigated the current state-of-the-art representations. In the following, we study our robust quantization method.

Table 1 presents the unit edit distance metric using our robustness method with and without the iterative approach. Compared with the k-means method, which is currently in use, our non-iterative method consistently outperforms it by a large margin (relative improvement of at least 30%). We also note that different augmentations affect the representation differently. Our iterative method provides a slight but consistent improvement over the non-iterative method. It is noticeable that the UED is increasing (i.e., worse performing) with the number of units used.

Table 3: Speech-to-Speech Translation results: We report BLEU scores for the proposed method and compare it against Lee et al. (2022). We report both development and test sets results. It is noteworthy that our approach allows for an improvement over the k-means used.

| Evaluation set | # units | Method | Spanish-English | French-English |
|---|---|---|---|---|
| Development | 500 | Ours | 17.3 | 16.4 |
| | 1000 | Ours | **18.2** | **17.5** |
| | 1000 | Lee et al. (2022) | 15.4 | 16.0 |
| Test | 500 | Ours | 14.4 | 15.75 |
| | 1000 | Ours | **15.9** | **17.1** |
| | 1000 | Lee et al. (2022) | 13.1 | 15.4 |

## 5.3 ENCODING METRICS

The ABX test examines the discriminative phonetic abilities of the representation. Versteegh et al. (2015) show that the ABX result is a good proxy to signal content (i.e., Phoneme Error Rate). The input to the ABX is a pair of words with a phoneme modification and a reference word containing the same phoneme as one of the pair's words. Next, the ABX measures the distance of the test phoneme representation to both the correct and incorrect representations. Finally, the distance between the test and the correct representation is expected to be lower than the distance to the incorrect representation. The ABX test is conducted in two setups: 'within' and 'across.' 'Within' is evaluated on input data from the same speaker, while 'across' is evaluated on input data from different speakers.

Table 2 shows the ABX results for both Librispeech 'clean' and 'other'. In our experiments, we found that the ABX score consistently and significantly improved on all the setups we tested. In this case, the iterative approach improves more than the non-iterative one, but the improvement is inconsistent. For a small number of units and the 'other' split, the ABX score is lower than the iterative model's score.

The spot-the-word test (WUGGY) requires detecting the real word from a pair of short utterances such as 'brick' vs. 'blick.' The detection is done by comparing the probabilities given by a language model for each word. This allows comparing representations by training language models on top of them. Differently, the acceptability judgment test (BLIMP) requires detecting the syntactically correct sentence from a pair of sentences, one of which is syntactically correct and the other is wrong. The detection is based on the perplexity of the language model. As presented in Table 2, our method enables improvement in all the investigated setups for both the spot-the-word and acceptability judgment tests. This is especially noticeable for a larger number of units. For instance, when considering 200 or 500 units, the absolute improvement of the WUGGY metric is 4.17 and 3.21, respectively.

## 5.4 SPEECH-TO-SPEECH TRANSLATION

Lee et al. (2022) propose a textless speech-to-speech translation method by forwarding a source speech signal and predicting its target's discrete representation. The authors use a k-means model trained on top of a multilingual HuBERT (mHuBERT) for speech representation. Additionally, the authors show that solving an auxiliary task enhances performance. We investigate the impact of using our robust quantizer as an alternative to the k-means used by Lee et al. (2022). Differently, we use HuBERT (instead of mHuBERT). Besides that, we follow the same setup in terms of model, computation resources, and data. To evaluate the quality of the translation, the authors use the sentence BLEU score (SacreBLEU) (Post, 2018).

Table 3 presents the results for the Spanish-English and French-English setups on the Europarl-ST development and test sets (Iranzo-Sánchez et al., 2020). It also shows the original result from Lee et al. (2022). The proposed method improves over Lee et al. (2022) under all the evaluated setups. Note, these results are especially interesting as the proposed method was trained on significantly less data (ours was trained on 1k hours while Lee et al. (2022) was trained on 100k hours).

## 6 RELATED WORK

In this work, we investigate the robustness of self-supervised representations for language modeling. This is related to the advancements in speech self-supervised learning, their robustness, and modern generative spoken language modeling. In the following, we review all three areas.

**Self-supervised learning.** The field of deep learning research has significantly benefited from self-supervised learning. Commonly, it involves encoding the input data and performing a task that enforces the representation to learn contextual embeddings. Speech self-supervised learning can be divided into two lines of research.

The first is discriminative, Oord et al. (2018) introduced Contrastive Predictive Coding (CPC), which trains a convolutional encoder and a predictor for future embeddings of the encoder using a contrastive loss. On top of it, Kharitonov et al. (2021b) propose to use time domain augmentations to improve the CPC model further. Wav2vec2 (Schneider et al., 2019) suggest using a contrastive loss that requires distinguishing between true and false future audio samples. Later, wav2vec2 (Baevski et al., 2020) learn quantized units using Gumbel softmax and predict masked spans of the latent speech representation. HuBERT (Hsu et al., 2021) employ a frame-based masked prediction task. First, it quantizes input frames and then predicts masked frames.

The second line of work is generative. An early generative self-supervised work is Autoregresstive Predictive Coding (Chung et al., 2019), which predicts the spectrum of a future frame. Later, Liu et al. (2020) introduced Mockingjay, which learns its representation by predicting non-causal context. TERA (Liu et al., 2021) alters time, frequency, and magnitude. Then it is required to reconstruct acoustic frames from altered versions.

**Robustness.** A desired property of a spoken language representation is robustness to augmentations that do not change the spoken information. The spoken information should not differ significantly when male and female speakers say the same content. There is an interesting trade-off between training a robust representation and the quality of the input data. It is possible, for example, to use the same speaker for all data points in the training set. The model would not be able to learn any speaker bias, but this constraint prevents scaling.

Recently, the robustness of self-supervised speech representations has gained attention from the community. WavLM (Chen et al., 2022) proposes adopting the well-known HuBERT model (Hsu et al., 2021) and training it with an additional denoising process. The authors apply a noising process to the training data and then predict the clean units from it. ContentVec (Qian et al., 2022) is focused on the disentanglement of a speaker from self-supervised speech representation. The authors propose to use three disentanglement components. First, the student network is disentangled through two transformations. Then the representations are forwarded through a speaker condition component. Finally, voice-converted input data points are used to generate teacher labels.

## 7 CONCLUSIONS

In this work, we first propose a metric for evaluating the robustness of self-supervised speech representations applied for spoken language modeling tasks. Equipped with the aforementioned metric, we point out the lack of robustness in current state-of-the-art speech encoders with respect to simple signal variations that do not alter the spoken information. We then propose a simple and effective method to boost the robustness of the current approaches and demonstrate it on three state-of-the-art self-supervised speech representation models. We empirically show the efficacy of the proposed approach when considering encoding methods together with a textless speech-to-speech translation task considering both Spanish-English and French-English translations.

As for broader impacts, this work is the first (to the best of our knowledge) which analyzes self-supervised speech representation models, considering basic signal variations. We hope that with the provided analysis and evaluation, researchers working on spoken language modeling and self-supervised speech representation learning will consider reporting the proposed metric setup along with evaluation of down stream tasks.

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

This appendix begins with a detailed explanation on the Levenshtein distance (Section A). Then, in Section B, we present additional results. We report results on two additional state-of-the-art self-supervised speech representations. We show that our method is indeed effective for those representations as well as shown in the main paper.

## A    LEVENSHTEIN DISTANCE

Throughout the paper, we use a version of the Levenshtein distance. In this section, we detail the Levenshtein distance between two sequences. Let $x \in \{1, .., K\}^{T_x}$ and $y \in \{1, .., K\}^{T_y}$ be two discrete vectors, not necessary in the same size. Let us also denote the operator $\text{tail}(x)$ to return a copy of the vector $x$ without its first element. Then, the Levenshtein distance is defined recursively by:

$$\text{Lev}(x, y) = \begin{cases} |x|, & \text{if } |y| = 0 \\ |y|, & \text{if } |x| = 0 \\ 1 + \min \begin{cases} \text{Lev}(\text{tail}(x), y) \\ \text{Lev}(x, \text{tail}(y)) \\ \text{Lev}(\text{tail}(x), \text{tail}(y)) \end{cases}, & \text{otherwise} \end{cases} \tag{4}$$

where $|x|, |y|$ are the lengths of the vectors $x$ and $y$ respectively. Note, in our implementation, we use deduplicated sequences.

## B    ADDITIONAL RESULTS

In the following, we provide additional results on the state-of-arts representations "wav2vec2" and "WavLM" (Baevski et al., 2020; Chen et al., 2022).

Tables 4 and 5 present the UED scores for both the wav2vec2 and WavLM models. Using our method, we observe robustness improvements for both of the models. However, it is notable that the WavLM model is more robust than the wav2vec2 model. It is reasonable since the WavLM trained to be a more robust model using noisy training samples.

Tables 6 and 7 present the discriminative and generative metrics for both wav2vec2 and WavLM. We observe a consistent improvement using our robust quantizer as in the robustness metrics. However, for the WavLM, the improvements are sometimes marginal (except for $k = 50$ where k-means outperforms our method). The WavLM model is trained with a HuBERT architecture, with more data and noisy samples. Interestingly, while presenting better performance on various downstream tasks than HuBERT, their ABX, WUGGY, and BLIMP scores are lower.

Table 4: Wav2vec2 unit edit distance

| # units | Method | Augmentation | | | |
|---|---|---|---|---|---|
| | | Time | Pitch shift | Reverberation | Noise |
| 50 | k-means | $50.81_{\pm 0.41}$ | $58.66_{\pm 1.16}$ | $43.71_{\pm 0.77}$ | $32.17_{\pm 0.61}$ |
| | Ours | $38.74_{\pm 0.45}$ | $42.33_{\pm 0.97}$ | $33.69_{\pm 0.73}$ | $25.36_{\pm 0.49}$ |
| | Ours (Iterative) | $\mathbf{36.68}_{\pm 0.39}$ | $\mathbf{40.29}_{\pm 1.04}$ | $\mathbf{33.28}_{\pm 0.74}$ | $\mathbf{23.99}_{\pm 0.51}$ |
| 100 | k-means | $55.30_{\pm 0.61}$ | $65.23_{\pm 0.91}$ | $48.41_{\pm 0.72}$ | $33.97_{\pm 0.46}$ |
| | Ours | $42.32_{\pm 0.46}$ | $47.07_{\pm 0.88}$ | $36.83_{\pm 0.71}$ | $27.15_{\pm 0.75}$ |
| | Ours (Iterative) | $\mathbf{40.43}_{\pm 0.57}$ | $\mathbf{45.73}_{\pm 0.90}$ | $\mathbf{36.34}_{\pm 0.77}$ | $\mathbf{26.22}_{\pm 0.59}$ |
| 200 | k-means | $59.85_{\pm 0.39}$ | $70.80_{\pm 1.31}$ | $53.13_{\pm 0.67}$ | $36.64_{\pm 0.62}$ |
| | Ours | $46.84_{\pm 0.42}$ | $51.60_{\pm 1.21}$ | $\mathbf{40.54}_{\pm 0.66}$ | $32.61_{\pm 0.67}$ |
| | Ours (Iterative) | $\mathbf{44.90}_{\pm 0.35}$ | $\mathbf{49.59}_{\pm 1.25}$ | $40.58_{\pm 0.62}$ | $\mathbf{29.49}_{\pm 0.57}$ |
| 500 | k-means | $66.12_{\pm 0.48}$ | $77.01_{\pm 0.98}$ | $59.69_{\pm 1.01}$ | $37.22_{\pm 0.65}$ |
| | Ours | $51.65_{\pm 0.49}$ | $\mathbf{55.40}_{\pm 1.03}$ | $45.85_{\pm 0.93}$ | $33.17_{\pm 0.62}$ |
| | Ours (Iterative) | $\mathbf{50.50}_{\pm 0.53}$ | $57.12_{\pm 1.02}$ | $\mathbf{44.67}_{\pm 0.98}$ | $\mathbf{31.92}_{\pm 0.69}$ |

Table 5: WavLM unit edit distance

| # units | Method | Augmentation | | | |
|---|---|---|---|---|---|
| | | Time | Pitch shift | Reverberation | Noise |
| 50 | k-means | $47.66_{\pm0.49}$ | $52.93_{\pm1.02}$ | $33.45_{\pm0.62}$ | $28.46_{\pm0.61}$ |
| | Ours | $39.12_{\pm0.43}$ | $44.25_{\pm1.06}$ | $31.58_{\pm0.62}$ | $25.32_{\pm0.67}$ |
| | Ours (Iterative) | $\mathbf{36.79}_{\pm0.46}$ | $\mathbf{40.16}_{\pm1.05}$ | $\mathbf{25.73}_{\pm0.64}$ | $\mathbf{25.01}_{\pm0.66}$ |
| 100 | k-means | $52.61_{\pm0.51}$ | $58.44_{\pm0.72}$ | $36.27_{\pm0.45}$ | $29.44_{\pm0.64}$ |
| | Ours | $43.55_{\pm0.53}$ | $49.03_{\pm0.75}$ | $30.54_{\pm0.44}$ | $25.93_{\pm0.67}$ |
| | Ours (Iterative) | $\mathbf{42.11}_{\pm0.50}$ | $\mathbf{46.08}_{\pm0.74}$ | $\mathbf{28.88}_{\pm0.47}$ | $\mathbf{25.47}_{\pm0.59}$ |
| 200 | k-means | $58.50_{\pm0.42}$ | $64.75_{\pm1.02}$ | $41.05_{\pm0.54}$ | $30.93_{\pm0.62}$ |
| | Ours | $49.57_{\pm0.41}$ | $53.48_{\pm1.09}$ | $34.29_{\pm0.53}$ | $26.66_{\pm0.65}$ |
| | Ours (Iterative) | $\mathbf{47.82}_{\pm0.46}$ | $\mathbf{52.47}_{\pm1.01}$ | $\mathbf{32.88}_{\pm0.55}$ | $\mathbf{26.09}_{\pm0.62}$ |
| 500 | k-means | $64.25_{\pm0.67}$ | $70.55_{\pm0.75}$ | $45.63_{\pm0.83}$ | $33.17_{\pm0.71}$ |
| | Ours | $55.41_{\pm0.64}$ | $59.79_{\pm0.87}$ | $42.85_{\pm0.78}$ | $28.46_{\pm0.79}$ |
| | Ours (Iterative) | $\mathbf{52.92}_{\pm0.69}$ | $\mathbf{57.840}_{\pm0.81}$ | $\mathbf{40.46}_{\pm0.81}$ | $\mathbf{27.09}_{\pm0.72}$ |

Table 6: Wav2vec2 discriminative and generative evaluation metrics.

| # units | Method | ABX (clean) ↓ | | ABX (other)↓ | | WUGGY ↑ | BLIMP ↑ |
|---|---|---|---|---|---|---|---|
| | | Within | Across | Within | Across | | |
| 50 | k-means | 12.03 | 15.31 | 13.61 | 19.07 | **49.76** | 53.92 |
| | Ours | 11.18 | 13.82 | 13.34 | 18.39 | - | - |
| | Ours (Iterative) | **10.35** | **12.75** | **12.64** | **17.29** | 49.65 | **55.29** |
| 100 | k-means | 11.27 | 13.99 | 13.06 | 17.11 | 51.63 | 53.87 |
| | Ours | 9.86 | 11.81 | 11.44 | 16.63 | - | |
| | Ours (Iterative) | **9.24** | **11.30** | **11.37** | **16.14** | **51.90** | **54.95** |
| 200 | k-means | 11.13 | 14.42 | 12.37 | 18.02 | 51.29 | 54.99 |
| | Ours | 10.19 | 12.41 | 11.85 | 17.52 | - | - |
| | Ours (Iterative) | **9.00** | **11.11** | **11.49** | **16.53** | **51.99** | **55.67** |
| 500 | k-means | 12.06 | 15.61 | 13.77 | 19.94 | 52.21 | 54.32 |
| | Ours | 10.76 | 13.83 | 13.52 | 19.60 | - | - |
| | Ours (Iterative) | **10.16** | **12.42** | **12.56** | **18.24** | **52.93** | **55.17** |

Table 7: WavLM discriminative and generative evaluation metrics.

| # units | Method | ABX (clean) ↓ | | ABX (other)↓ | | WUGGY ↑ | BLIMP ↑ |
|---|---|---|---|---|---|---|---|
| | | Within | Across | Within | Across | | |
| 50 | k-means | 7.60 | 9.06 | **9.22** | 12.99 | 63.91 | 55.29 |
| | Ours | 7.41 | 8.68 | 9.51 | **11.78** | - | - |
| | Ours (Iterative) | **7.19** | **8.25** | 9.41 | 11.87 | **64.87** | **55.81** |
| 100 | k-means | 6.91 | 8.06 | 8.95 | 11.86 | 63.61 | 54.59 |
| | Ours | **6.02** | 7.13 | 8.36 | **10.95** | - | - |
| | Ours (Iterative) | 6.39 | **7.02** | **8.17** | 11.21 | **63.99** | **54.97** |
| 200 | k-means | 6.74 | 8.12 | 8.76 | 12.09 | 65.97 | 55.59 |
| | Ours | **6.40** | **7.45** | **8.61** | **11.49** | - | - |
| | Ours (Iterative) | 6.51 | 7.73 | 8.93 | 11.94 | **66.90** | **55.89** |
| 500 | k-means | 7.14 | 8.10 | 9.09 | 11.70 | 64.56 | 55.91 |
| | Ours | **7.03** | **7.91** | **8.99** | **11.21** | - | - |
| | Ours (Iterative) | 7.08 | 7.87 | 9.03 | 11.54 | **65.81** | **56.09** |

