# OpenReview forum: "On the robustness of self-supervised models for generative spoken language modeling"
_ICLR.cc/2023/Conference — Submitted to ICLR 2023_

### Official Review · Reviewer_PUau · 2022-10-23

**Confidence:** 3
**Correctness:** 3
**Technical Novelty And Significance:** 2
**Empirical Novelty And Significance:** 3
**Recommendation:** 6

**Clarity, Quality, Novelty And Reproducibility:**

The paper is clearly written for the most part and doesn’t have significant quality issues. The paper has moderate novelty to the best of my knowledge. It is not clear if the authors have or intend to release their code for reproducibility.


**Strength And Weaknesses:**

Strengths:

The paper is written clearly and is easy to follow. The proposed techniques themselves are simple yet effective for getting better discrete representations from speech. Evaluation is done on a variety of tasks and a significant improvement is achieved on all of them.

Weaknesses:

My main concern about this paper is how the premise is set. The main premise in this paper seems to be that spoken language modeling is sensitive to noise and discretized speech representations are not robust to it. My question is how often is noise a problem in speech generation/synthesis for downstream tasks like end-to-end speech translation. Maybe I am missing something but it seems that speech processing for spoken LM is not subject to noise from outside as it processes its own generated speech and not human speech. Also, was there a source of noise in tasks that involve autoregressive generation like E2E S2S translation?

Maybe I am missing something here, but it seems that the UED score analysis does not add anything noteworthy to the paper. I understand that it shows how a discrete representation of speech may change in the presence of noise but it’s not very surprising that it changes as representations always change in the presence of noise. The question is how much is the downstream task affected by these noises. There are no experiments which show the effect of artificial noise on downstream tasks but only UED is used to show the effect which is rather obvious. The fact that UED improves after a teacher-student like training with CTC loss is also not surprising. If a UED score is high, does it necessarily mean that one representation is better than the other?


**Summary Of The Paper:**

This paper first studies how presence of different kinds of noise affect the discrete speech representations from self-supervised models used for spoken LM. The authors utilize the Levenstien distance between sequences of discrete token IDs obtained from HuBERT as an indicator for the presence of noise. Then, a pseudo-labelling task is introduced that aims to reduce this noise. The proposed techniques improve model performance on a variety of tasks.

**Summary Of The Review:**

I found the CTC based pseudo-labelling and its iterative counterpart to be quite ingenious and interesting. It improves the overall discrete representation quality of the speech through a teacher-student like framework. But I do question how the story is set up by showing that representations change by addition of synthetic noise through the use of UED. I found that part rather redundant. Maybe the authors can convince me as to why it’s important to this paper.

---

> ### Author Response · Authors · 2022-11-13
> **Response to Reviewer PUau**
>
> **Re. Noise problem in noisy representations:** Augmentations such as background noise, speech variability, and different speakers may be significant issues when considering real-world data. For instance, the same person says the same sentence twice. As not two speech recordings are identical, minor time stretches might happen (e.g., saying specific syllables shorter / longer). The paper shows that this can result in a significant difference between the representations (~40%). Such a phenomenon makes modeling and generation more challenging. Another example would be dev-other and test-other from the Librispeech corpus (results are reported in Table 2), which contains reverberant accented speech. In which the proposed method significantly and consistently improves over the standard k-means.
>
> **Re. Speech processing for spoken LM is not subject to noise:** This is true only during inference time. During training time, the model learns from real-world examples which are noisy in nature. Please note that when we refer to noisy samples, we consider more than just background noise. Specifically, we consider reverberant signals, unseen speakers, and pronunciation variations.
>
> **Re. Representation change:**  We agree that it is not surprising. However, the level of change is surprising. Additionally, we consider more than just background noise. Specifically, we consider time-stretch to simulate pronunciation variations, pitch-shift to simulate speaker variations, and lastly, background noise and reverberation to simulate in-the-wild recordings.
>
> Generally, when operating over discrete representations, we ideally would like the representation to be as consistent as possible when considering different signal variations. For example, think of the extreme case of text. A text-based speech translation would first run an ASR to convert the speech into text. The text representation should be invariant to time-stretch / pitch-shift / etc. In this study, we aim to get to the same level of robustness using "pseudo-text,” i.e., our discrete representations.
>
> **Re. UED high equals better performance?:** This is a great question. We extensively studied it in the paper. Low UED does not directly mean better performance. For example, a quantizer with only one cluster will obtain a perfect UED but will not be able to learn. Hence, we believe the UED is a complementary metric to the current ones, and a set of metrics should be considered.
>
> **Re. Analysis redundant?:** Recent studies on generative spoken language modeling are based on k-means representation computed over SSL models [1-7]. However, such methods usually operate over academic benchmarks, which are relatively clean and constructed under controlled settings. When considering noisy conditions, such representations vary a lot, which makes it hard to model it properly (for example, think of the same sentence spoken by two different persons getting significantly different token sequences). To study such phenomena under controlled settings, we define a set of four different augmentations (time-stretch, pitch-shift, additive noise, reverberation) and evaluate the robustness of such representations. To quantify the robustness of the representations, we propose using the UED over clean and augmented data. By that, we are the first to reveal and measure the lack of robustness in such representations.
>
> **Re. Code release:** Upon acceptance, we will release the code for our method and metric. We will also release converged quantizers for a variety of unit numbers.
>
> [1] Lakhotia, Kushal, et al. "On generative spoken language modeling from raw audio." Transactions of the Association for Computational Linguistics 9 (2021). \
> [2] Kharitonov, Eugene, et al. "Text-free prosody-aware generative spoken language modeling." (2021). \
> [3] Nguyen, Tu Anh, et al. "Generative Spoken Dialogue Language Modeling." (2022). \
> [4] Borsos, Zalán, et al. "Audiolm: a language modeling approach to audio generation." (2022). \
> [5] Qian, Kaizhi, et al. "ContentVec: An improved self-supervised speech representation by disentangling speakers." International Conference on Machine Learning. PMLR, 2022. \
> [6] van Niekerk, Benjamin, et al. "Analyzing speaker information in self-supervised models to improve zero-resource speech processing." (2021). \
> [7] Dunbar, Ewan, et al. "The zero resource speech challenge 2021: Spoken language modeling." (2021).

---

> > ### Comment · Reviewer_PUau · 2022-11-22
> > **Further comments**
> >
> > I thanks the authors for their response.
> >
> > I still believe that the paper has some redundancy issues in that the UED results are not surprising. The authors say that the level of change is surprising but what is the baseline they are comparing against?
> >
> > Also, the authors say that representation change may be a problem is noisy scenarios, yet the only result they provide under noisy scenarios are the UED scores. What happens to the downstream tasks like S2S translation using a noisy dataset or even using synthetic noise? How much does the performance deteriorate and how much of it is recovered using the proposed technique?
> >
> > In the end, I still believe this paper is well suited as a somewhat novel use of the teacher-student technique to improve representation. In light of this, I will keep my score to 6.

---

### Official Review · Reviewer_wx94 · 2022-10-24

**Confidence:** 3
**Correctness:** 2
**Technical Novelty And Significance:** 2
**Empirical Novelty And Significance:** 2
**Recommendation:** 5

**Clarity, Quality, Novelty And Reproducibility:**

UED definition 3.1: Where exactly is the deduplication in there? Function E outputs the same length as the input?

Figure 2, UED is in percentage? Or otherwise, shouldn't it be in the range 0 and 1?

Speech-to-speech translation results: I don't really see how they directly test the influence of their method. They just compare it to some result from the literature. But they should have a baseline without the proposed method, and then do a comparison with the proposed method, to directly see the difference.

Pseudo-labeling: It's not really clear to me: What are the targets for CTC? Those quantization indices, these found units? I think this should be made more clear. Further, this depends on a previous pretrained model?


**Strength And Weaknesses:**


Strength:

- Tests on time-stretch, pitch-shift, additive-noise, and reverberation and how they affect the learned representation.
- Pseudo-labeling using CTC, and an iterative variant improve speech-to-speech translation.

Weaknesses:

- The paper claims to study the self-supervised representation in general for spoken language modeling. However, in the end it really only tests it for speech-to-speech translation. This limits the scope of the paper. I would have expected tests for other downstream tasks as well, to really get an understanding on the robustness of self-supervised representation, as the title says.
- The actual tests on robustness are very limited. It just has a single experiment on speech-to-speech translation. I would have expected more experiments to verify the robustness aspect. Note that this is a separate weak point to the previous. Here in this point, I specifically mean that the robustness experiments are too limited. Of course, both are related. When experiments are done on other tasks as well, this would automatically extend also the actual tests on robustness. But there are potential other ways to extend such tests as well.


**Summary Of The Paper:**

Setting: self-supervised representations for speech (spoken language modeling)

Tasks: speech-to-speech translation

The paper wants to improve self-supervised representations for speech.

Tests how variations of time-stretch, pitch-shift, additive-noise, and reverberation alter the learned representations. It should not really affect them, as those properties should not really be relevant, at least for most downstream tasks.

A new method using pseudo-labeling and CTC training is introduced to improve the robustness.


**Summary Of The Review:**

I think the scope of the work is too limited. When studying the learned representation, other downstream tasks should be tested, which would test the robustness much more directly, such as speech recognition.

---

> ### Author Response · Authors · 2022-11-13
> **Response to Reviewer wx94**
>
> **Re. Experiments:** Please note that we report more than the speech-to-speech translation metric. We also report the ABX, WUGGY, and BLIMP metrics. Those metrics are widely used and are the foundation of generative spoken language modeling and zero resource speech evaluation setup, see [1, 4, 7] for example. These metrics evaluate the quality of the speech representation considering linguistic content, syntactic and semantic structure. Those metrics give a better understanding of the robustness of our method since they investigate specific aspects of the representation.
>
> **Re. Definition 3.1:** For notation clarity, we kept the deduplication only in the text. We further bolded it in the revision.
>
> **Re. function E:** The function E output is the same length as the number of frames in the input.
>
> **Re. Figure 2:** We report the scores multiplied by a hundred (mentioned in Table 1 caption). We clarified it in the revision.
>
> **Re. Speech-to-speech results:** We used a state-of-the-art speech-to-speech translation from the literature that uses k-means (i.e., our baseline). In our evaluation setup, the only change is that we use our quantization methods instead of the k-means. In other words, the results in Table 3 compare the same base model where we only modify the quantization method (k-means vs. the proposed method). Considering that, we think this is exactly the reviewer’s intention.
>
> **Re. Pseudo-labeling:** The targets are indeed cluster indices. Can the reviewer clarify what they refer to as a “previous pre-trained model”?
>
> **Re. Scope of the work:** Please note that the proposed method is applied over quantized representation and mainly aims at generative spoken language modeling over discrete representations. Such research direction is emerging recently and shows great potential [1-7], to name a few.
>
> We do not claim to improve ASR metrics or any other discriminative speech task (e.g., speaker classification, spoken term detection, etc.), as our goal in this study is to improve the robustness of speech discrete representations obtained from self-supervised models, so later on, applying generative and modeling tasks over it yields better performance and more consistent representation. Hence, we consider zero-speech challenge metrics and generative spoken language modeling metrics (i.e., ABX, WUGGY, BLIMP) together with textless speech-to-speech translation operating over discrete representations. After demonstrating that current discrete representations are sensitive to basic signal variations which do not alter the spoken content, we proposed a method to mitigate that and demonstrate that under all settings, the proposed approach is superior to the baseline methods. Due to all of the above, we believe speech recognition, or any other discriminative task, is not a good fit for such a research area.
>
> [1] Lakhotia, Kushal, et al. "On generative spoken language modeling from raw audio." Transactions of the Association for Computational Linguistics 9 (2021). \
> [2] Kharitonov, Eugene, et al. "Text-free prosody-aware generative spoken language modeling." (2021). \
> [3] Nguyen, Tu Anh, et al. "Generative Spoken Dialogue Language Modeling." (2022). \
> [4] Borsos, Zalán, et al. "Audiolm: a language modeling approach to audio generation." (2022). \
> [5] Qian, Kaizhi, et al. "ContentVec: An improved self-supervised speech representation by disentangling speakers." International Conference on Machine Learning. PMLR, 2022. \
> [6] van Niekerk, Benjamin, et al. "Analyzing speaker information in self-supervised models to improve zero-resource speech processing." (2021). \
> [7] Dunbar, Ewan, et al. "The zero resource speech challenge 2021: Spoken language modeling." (2021).

---

> > ### Comment · Reviewer_wx94 · 2022-11-30
> > **Comment**
> >
> > > We do not claim to improve ASR metrics or any other discriminative speech task,  as our goal in this study is to improve the robustness of speech discrete representations.
> >
> > I don't understand this. When improving the robustness of the speech representation, some discriminative task like ASR seems to be a much better indirect measure of the improved robustness than speech-to-speech translation?

---

> > > ### Author Response · Authors · 2022-12-05
> > > **Response**
> > >
> > > We agree with the reviewer that discriminative tasks should improve upon an improvement of representation. However, the focus of our work is on robust discrete representations for generative models. Therefore, our method operates on discrete spaces. Discriminative tasks (such as ASR) do not usually operate on discrete representations, hence we believe this is out of scope for this paper. Please also note that we do not modify the original HuBERT continuous representation.
> > >
> > > Nevertheless, we do assess the representation's ABX scores, which examine its discriminative phonetic capabilities, which is a standard metric for these tasks (see [1-3]). In Table 2, we show that we consistently improve this score on both 'clean' and 'other' (noisy) Librispeech evaluation setups.
> > >
> > > [1] Borsos, Zalán, et al. "Audiolm: a language modeling approach to audio generation." (2022). https://arxiv.org/abs/2209.03143 \
> > > [2] Lakhotia, Kushal, et al. "Generative spoken language modeling from raw audio." Transactions of the Association for Computational Linguistics 9 (2021). https://arxiv.org/abs/2102.01192 \
> > > [3] Dunbar, Ewan, et al. "The zero resource speech challenge 2021: Spoken language modeling." (2021). https://arxiv.org/abs/2104.14700

---

> > > > ### Comment · Reviewer_wx94 · 2022-12-05
> > > > **Comment**
> > > >
> > > > I don't understand why it matters whether a representation is discrete or continuous. Why is this relevant for anything what I argued? I don't think this is relevant.

---

> > > > > ### Author Response · Authors · 2022-12-08
> > > > > **Response**
> > > > >
> > > > > We are not arguing that the reviewer’s suggestion is wrong.
> > > > >
> > > > > We are stating that our paper investigates robust representations for generative spoken language modeling. The ASR task is discriminative and hence we are not evaluating it. However, we do perform the common evaluation for discriminative tasks and show it has also been improved.
> > > > >
> > > > > Our method might be effective for other tasks as well, but this is not in our paper's scope. We would be happy to see future work use our method in different directions.

---

> > ### Comment · Reviewer_wx94 · 2022-11-30
> > **Comment**
> >
> > > Re. Pseudo-labeling: The targets are indeed cluster indices. Can the reviewer clarify what they refer to as a “previous pre-trained model”?
> >
> > Ok then I misunderstood this part. I thought the targets came from the recognition output of another model.

---

### Official Review · Reviewer_EDiE · 2022-10-24

**Confidence:** 5
**Clarity, Quality, Novelty And Reproducibility:** The article is easy to follow, but th…
**Correctness:** 4
**Technical Novelty And Significance:** 2
**Empirical Novelty And Significance:** 2
**Recommendation:** 3

**Strength And Weaknesses:**

Strength: the approach of using augmentation robustness as an objective for improving speech representation in self-supervised learning is well-motivated.
Weaknesses: both the noisy-student learning and the quantization distillation are existing approaches, and the idea of correlating semantics with noise robustness is also well known. This makes the novelty of the work limited. The experiments on downstream tasks are also insufficient to demonstrate the benefits of the approach.


**Summary Of The Paper:**

This work looks into capturing the semantic information of self-supervised learning for speech. In particular the central idea is that the semantic of the speech should be robust to the small amount of noise in the speech, and therefore the robustness of a self-supervised learning algorithm toward noise should correlate with how well its learned representation captures the semantic of the speech. The approach uses speech augmentation to study the robustness of existing self-supervised learning algorithms, and proposes to apply a noisy-student approach with CTC loss to distill the quantizer. The experiments show the distillation approach provides a better quantization based on ABS test, spot-the-word test, and the acceptability judgment test, and there is some improvement on a downstream speech-to-speech translation task.

**Summary Of The Review:**

This work demonstrates that by applying noisy-student training on quantizers learned from self-supervised learning, the resulting student quantizer can provide some improvement. Overall the novelty of the work is limited, and there should be more experiments to demonstrate the benefit of the approach.

---

> ### Author Response · Authors · 2022-11-13
> **Response to Reviewer EDiE**
>
> We kindly disagree with the reviewer. We believe that using a simple and effective technique to improve a major flaw in an established domain should not be considered a limitation but rather an advantage. To better highlight our contribution: (i) we are the first to point out the described issue with current self-supervised representations applied to generative spoken language modeling; (ii) next, we proposed a metric to evaluate it and, using it, we analyzed various state-of-the-art representations; (iii) lastly, we proposed **a simple and effective method** to improve the proposed metric and demonstrate the effectiveness of such an approach on a broad set of evaluation metrics for spoken language modeling and speech to speech translation empirically.
>
> We believe that the community should be aware of such flaws, metrics to evaluate them, and a method to improve the robustness of the commonly used methods. **In light of the above, we believe this submission is novel and publication worthy.**
>
> Other than that, we would be happy to address any concrete concerns or feedback raised by the reviewer.

---

> > ### Comment · Reviewer_EDiE · 2022-12-09
> > **Response to authors**
> >
> > I agree with the authors that a simple approach would be highly appreciated if it is effective for addressing the issue. The experiments in the work, however, are too limited and do not show that the approach is sufficiently effective. Therefore, the work has limited contribution, and I would maintain my original recommendation.

---

### Official Review · Reviewer_TaQk · 2022-10-25

**Confidence:** 3
**Correctness:** 3
**Technical Novelty And Significance:** 2
**Empirical Novelty And Significance:** 3
**Recommendation:** 5

**Clarity, Quality, Novelty And Reproducibility:**

Overall, this paper clearly presents the proposed methodology. This work is reproducible with information presented in the paper. It is mostly based on existing technologies, with limited originality.

**Strength And Weaknesses:**

Strength: Overall, this propose approach is technical sound. Experiments and analysis are carefully designed for validation, including several widely used self-supervised learning representations.

Weakness: This proposed metric and approach are mostly based on existing technologies (Levenshtein distance, MLP-based module, CTC loss etc). Also I think more details are needed for explaining design and results. (See questions in the summary section below).

**Summary Of The Paper:**

In this paper authors propose the unit edit distance to measure robustness of self-supervised speech representations for spoken language modeling, and based on it, adding a multi-layer perceptron (MLP) trained using CTC to improve model robustness. Experimental results based on multiple self-supervised learning methods show effectiveness of the proposed approach.

**Summary Of The Review:**

Overall, the methodology presented in this paper is technical sound, and the experiments are well designed to show its effectiveness. Several widely used self-supervised learning techniques are considered. This proposed approach is mostly based on existing technics (Levenshtein distance, MLP, CTC etc), with limited originality. Also I'd suggest authors consider addressing the following questions:

1. For the proposed unit edit distance, how to address class permutation (between the two inputs) when computing the Levenshtein distance? Also please double check if dimensions match for the composition of E, f and g in formula (1).
2. As said in paper, the UED monotonically increases with the number of units used. Are there thoughts on how to define UED more robust re the number of units?
3. In Section 4.1, it's said "learning the parameters of the encoder performs worse than freezing them." Is there a hypothesis on why this is the case?
4. Why are std only added for those results in Table 1, but not 2 & 3?
5. For "Our quantizer is composed of three fully connected layers" mentioned in Section 5.1, what are the layers' sizes?

---

> ### Author Response · Authors · 2022-11-13
> **Response to Reviewer TaQk**
>
> **Re. Using existing techniques:** We kindly disagree with the reviewer. We believe that using a simple and effective technique to improve a major flaw in an established domain should not be considered a limitation but rather an advantage. To better highlight our contribution: (i) we are the first to point out the described issue with current self-supervised representations applied to generative spoken language modeling; (ii) next, we proposed a metric to evaluate it and, using it, we analyzed various state-of-the-art representations; (iii) lastly, we proposed a simple and effective method to improve the proposed metric and demonstrate the effectiveness of such an approach on a broad set of evaluation metrics for spoken language modeling and speech to speech translation empirically.
>
> **Re. Class permutation:** Our method aims to learn a more robust representation using augmentations that do not alter the spoken content. Please also note that our method is unsupervised, meaning we first extract pseudo-labels. Next, we perform a signal transformation, and lastly, we optimize the model to predict the pseudo-labels under augmentations. Considering the above, can the reviewer clarify the meaning of “class”?
>
>
> **Re. Number of units:** This is an interesting point. We associate this with the fact that with higher dimensions, there is a higher possibility of redundancy or relatively small and sensitive clusters. A low dimension plays a role of a regularizer for the robustness of the clusters, but it is not expressive enough to yield good generative performance. Hence, we believe that the UED should not be changed w.r.t this observation, but the discrete representation should be more robust, as we showed in our method.
>
> **Re. Training the upstream parameters:** Training the upstream’s parameters indeed pushes the UED scores even lower. However, zero-shot metrics, such as ABX, WUGGY, and BLIMP, perform worse. We find that freezing the upstream model is a suitable regularization. We clarified it in the revised version.
>
> **Re. Standard deviations:** Table 1 presents the UED distances on stochastic validation sets (i.e., augmentations for each sample are drawn with a predetermined probability). Hence we present the STD. Tables 2 and 3 present results using a single converged quantizer (k-means vs. ours). Furthermore, different quantization training seeds are computationally heavy for WUGGY and BLIMP metrics as they require retraining an additional language model for each setting.
>
> **Re. Layer sizes:** We use the division floor of the difference between the upstream dimension (768 for the base and 1024 for the big model) to the number of units. For example, for a base model and k=200:
>
> 768,  (768 - 200) // 2 \
> (768 - 200) // 2, (768 - 200) // 4 \
> (768 - 200) // 4, 200
>
> Thanks for pointing this out. We added this information to the revision.

---

> > ### Comment · Reviewer_TaQk · 2022-12-11
> > **Response to authors**
> >
> > Thanks for the authors' response and for sharing additional thoughts and details. This helps clarify some of my questions. However, I'm still not fully convinced the proposed methodology and experiments are strong enough for ICLR  (e.g. in general there are ways to derive std, the inherent monotonically increase with number of units etc) , so I'd keep my original scores.
> >
> > By "class" I mean the quantizer outputs (e.g. index of clusters from k-means). Thanks for the clarification.

---

### Author Response · Authors · 2022-11-18
**Remaining concerns**

We would like to thank again the reviewers for taking the time to review our paper. As we are approaching the end of the authors discussion, we would be happy to address any remaining concerns per the reviewer's request.

---

### Decision · Program_Chairs · 2023-01-20

**Decision:**

Reject

**Justification For Why Not Higher Score:**

Limited novelty and experiments.

**Justification For Why Not Lower Score:**

N/A

**Metareview: Summary, Strengths And Weaknesses:**

The paper studies the robustness capabilities of self-supervised speech representations for generative tasks, in particular spoken generative language models. The authors present a method for a teacher-student like training with CTC loss, and a metric for robustness evaluation. Experiments are conducted on a speech-to-speech translation task. The paper is clearly written and easy to follow. At the same time, the paper is considered by the reviewers as having moderate novelty and limited experiments.